# A Novel Dental Caries Model Replacing, Refining, and Reducing Animal Sacrifice

**Amit Wolfoviz-Zilberman** [1,2], **Yael Houri-Haddad** [1,†] **and Nurit Beyth** [1,*,†]

1   Department of Prosthodontics, Hadassah Medical Center, Faculty of Dental Medicine, Hebrew University of Jerusalem, Jerusalem 9112001, Israel; amit.wolfoviz@mail.huji.ac.il (A.W.-Z.); yaelho@ekmd.huji.ac.il (Y.H.-H.)
2   Institute of Dental Sciences, Hadassah Medical Center, Faculty of Dental Medicine, Hebrew University of Jerusalem, Jerusalem 9112001, Israel
*   Correspondence: Nurit.Beyth@mail.huji.ac.il
†   These authors contributed equally to this work.

**Abstract:** In vitro and in vivo models simulating the dental caries process enable the evaluation of anti-caries modalities for prevention and treatment. Animal experimentation remains important for improving human and animal health. Nonetheless, reducing animal sacrifice for research is desirable. The aim of the study was to establish a new reproducible in vitro caries model system and compare it to an in vivo model using similar conditions. Hemi-mandibles were extracted from previously euthanized healthy 10-week-old BALB/C female mice. Jaws were subjected to saliva, high-sucrose diet, and dental caries bacteria *Streptococcus mutans UA159* for 5 days. Similar caries induction protocol was used in vivo in fifteen BALB/c female mice (6–7 weeks old) and compared to the in vitro model. Caries lesions were assessed clinically by photographic analysis and μCT analysis, and bacterial growth was evaluated. Under in vitro experimental conditions, carious lesions evolved within 5 days, prominently in the depth of the occlusal fissures in the control group as depicted by photographic analysis, μCT analysis, and bacterial growth. The developed in vitro caries model presented in this study may be a novel animal sparing model for caries disease studies and can be used widely to evaluate the efficacy of different antibacterial dental materials.

**Keywords:** caries model; *Streptococcus mutans*; demineralization; dental caries



## 1. Introduction

Dental caries is a common infectious and transmissible disease. The basic mechanism is the demineralization of enamel and dentin via acid generated by bacterial biofilm. Oral bacteria metabolize carbohydrates, producing organic acid that diffuses into the enamel and dentin and dissolves minerals [1].

Although several studies have related the participation of other bacteria in the pathogenesis of dental caries [2–4], *Streptococcus mutans* plays a central role in the development of cariogenic biofilms [5]. The acidogenic and aciduric (associated with acid tolerance) properties of *S. mutans*, together with its ability to synthesize extracellular glucans, are the major factors for the development and establishment of cariogenic biofilms.

Caries models are valuable when trying to understand the complex process of the disease by addressing various questions related to biofilm formation, the process of caries development, and its prevention. Moreover, caries models are extremely valuable to determine prevention and treatment approaches prior to clinical application.

Various in vitro and in vivo techniques have been developed for caries simulation. Many of these studies focus on the evaluation of bacterial growth or biofilm composition rather than demineralization of the enamel [6–9]. Furthermore, most in vivo caries studies have been conducted using rats rather than mice and facilitated cariogenic bacteria by including oral shifting via desalivation and/or antibiotic treatment, which created a non-natural environment [10–15].

The rodent model has several advantages in mimicking human caries due to its similarity in genetics and hard tissues composition. Moreover, rodent caries models allow for a similar but accelerated caries development process: rodents have thinner layers of enamel and dentin, which may enhance the demineralization process. Previous in vivo studies in mice required 5–7 weeks of cariogenic bacteria inoculation for caries lesions development [10,16,17].

Animal experiments contribute essential knowledge to the medical and pharmaceutical research that cannot be achieved in vitro. Unfortunately, this knowledge requires millions of lab animals annually. The pain and death resulting from these experiments has been at the center of debate for a long time. Reducing animal sacrifice by developing alternative models without compromising the obtained knowledge is indeed required [18,19].

The objective of this study was to develop a new quantitative and reproducible in vitro caries model, using mice jaws of euthanized healthy mice that were sacrificed for other research purposes and are no longer used (for example control group mice). An in vivo caries model was designed to evaluate and verify the in vitro model's similarity and resemblance to the real oral conditions.

Preferably the combination of advantages of in vitro models with animal models may be beneficial for studying the caries disease. The ability to mimic the oral conditions in a controlled in vitro study using hard tissues that usually require a living host to model caries but without the animal sacrifice may be advantageous.

The null hypotheses of this study were that the development of an in vitro caries model using extracted jaws from euthanized healthy mice will mimic the common in vivo caries model and will introduce an alternative approach for dental research experiments.

## 2. Materials and Methods

### 2.1. Ethics Approval

Saliva was collected from 5 healthy volunteers, age range 22–67 (3 female, 2 men) (authorized by the institutional ethics committee #HMO-0706-16). All participants gave verbal and written assent to participate.

All animal experimental procedures were reviewed and approved by the IACUC of the Hadassah-Hebrew University Medical Center (MD-17-15315-3).

### 2.2. Bacterial Strains and Culture Conditions

*Streptococcus mutans UA159* was grown overnight in brain–heart infusion (BHI) broth (Difco, Detroit, MI, USA) at 37 °C. Bacterial suspensions adjusted to total bacterial load of $10^8$ CFU/mL. Viable counts using serial dilutions were used to assess bacterial concentration (CFU/mL).

### 2.3. Extracted Jaws Preparation

Ten sacrificed 10-week-old BALB/C female mice hemi-mandibles were obtained from previously euthanized healthy mice.

### 2.4. Saliva Sterilization

To simulate the oral cavity conditions, the samples were subjected to sterilized saliva according to Kalfas and Rnudegren [20] with a modification. Saliva was collected from healthy volunteers (authorized by the institutional ethics committee #HMO-0706-16). Dithiothreitol (DTT) 1M (Merck KGaA, Darmstadt, Germany) was added to saliva to reach 2.5 mM concentration. The saliva was cooled at 4 °C for 10 mins and then centrifuged for 15 mins. The upper fluid was discarded, and sterile double-distilled water (DDW) was added, adjusting to 25% saliva concentration. At last, the saliva was filtered using 0.22 µm filter and frozen for re-use.

### 2.5. Caries-Promoting Environment

Each sample was placed in a well in a 48-well flat-bottom microtiter plate (Nunclon, Nunc; Thermo Fisher Scientific Inc, Waltham, MA, USA) using a sterile tweezer. The samples were subjected to 50 μL sterilized saliva for 30 min. Then, 50 μL of bacterial suspension and 400 μL of brain–heart infusion (BHI) broth (Difco, Detroit, MI, USA) containing 5% sucrose and bacitracin (2.77 mg/mL) were added. BHI was replaced every 24 h for 5 days, after which the samples were analyzed. Five jaws were subjected to saliva, and BHI served as a control group. Figure 1 summarize schematically the in vitro caries model experiment course.

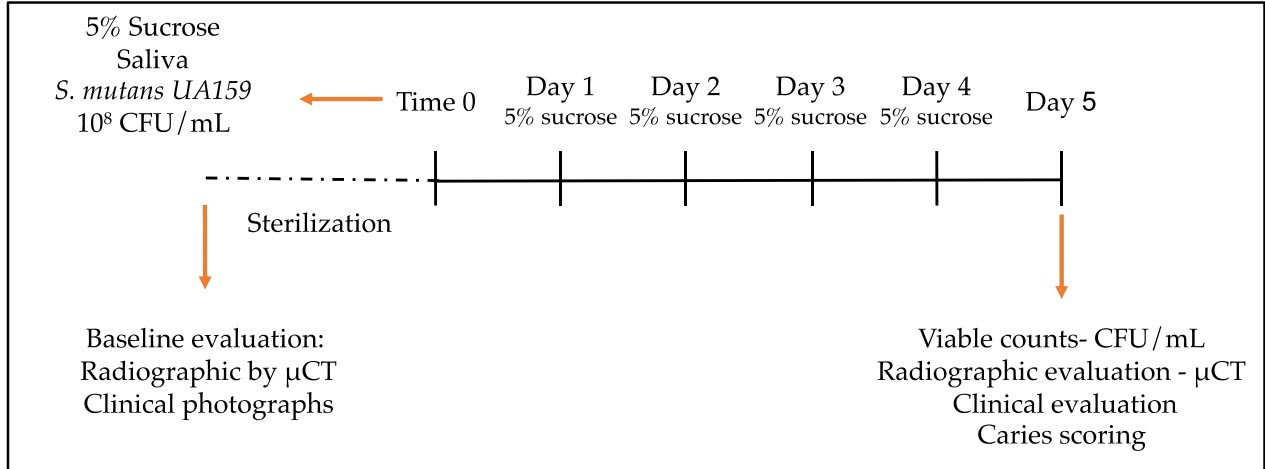

**Figure 1. Schematic diagram summarized in vitro caries model over extracted jaws from healthy euthanized mice**. Before subjecting the jaws to a caries-promoting environment, all jaws were photographed using an SMZ25 stereomicroscope (Nikon, Tokyo, Japan) and scanned using micro-computed tomography (μCT) (μCT 40, Scanco Medical, Switzerland). Then, all jaws were autoclaved. At time 0, jaws were subjected to saliva, bacterial infection by *S. mutans UA159*, and 5% sucrose growth media. Growth media was replaced every 24 h. A group of five jaws served as a control group without bacterial infection. After 5 days, microbiological samples were collected using a low-speed dental bur 0.25 mm (MDT, Israel) for viable counts evaluation. Then, hemi-mandibles were photographed again using SMZ25 stereomicroscope and scanned using μCT for demineralization quantification.

### 2.6. Photographic Depiction

Each jaw was photographed using an SMZ25 stereoscope (Nikon, Tokyo, Japan) before and after subjecting to caries-promoting conditions. A custom-made silicone stand was prepared for each jaw allowing a reproducible photographic capture taken in the similar conditions of light and resolution.

### 2.7. Carious Lesions Micro-Computed Tomography (MCT) Evaluation

Jaws underwent micro-computed tomography (μCT) (μCT 40, Scanco Medical, Switzerland) before and after experiment for demineralization quantification. Each jaw was placed in an Eppendorf tube containing 100 μL of phosphate-buffered saline (PBS) and autoclaved before experiment.

For quantitative 3-dimensional analysis of the mineral tissue loss, the hemi-mandibles were examined by a desktop μCT system (μCT 40, Scanco Medical, Switzerland) energy of 70 kV, intensity of 114 μA, and resolution of 6 $\mu m^3$ voxel size. Samples were placed in a cylindrical sample holder, and about 300 microtomographic slices were acquired covering the entire crown volume of each hemi-jaw. Sample scans, before and after experiment, were aligned as described by Goldman et al. [21]. In brief, three points of alignment in the first molar were chosen: the mesial apical constriction, the coronal part of the mesial canal, and the distal apical constriction. After alignment, a slice in which the coronal pulp, radicular pulp of both canals, and both apical foramens are presented was chosen (as demonstrated in

Figure 2). The total crown volume of first molar was marked as a rectangle from the mesial cemento-enamel junction and adjusted accordingly to all slices in which the crown's enamel was observed. Then, the marked total cubic volume was evaluated using a dedicated algorithm that gives the percentage of total volume for 6 selected ranges of density (by mgHA/ccm). Air and pulp have a similar radiographic density, which was examined and determined to 0–500 mgHA/ccm. The remaining five ranges have been standardized with respect to total volume. The caries density range is determined to 500–1500 mgHA/ccm, while healthy dentin and enamel were determined to 1500–3000 mgHA/ccm according to the respected density determined by the microCT machine.

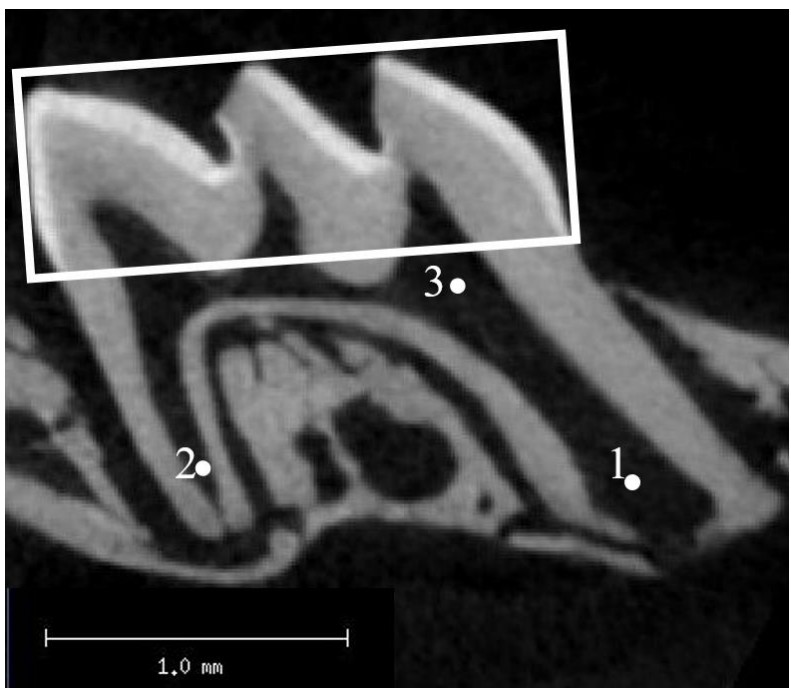

**Figure 2. Standardized μCT analysis.** μCT analysis included standardized alignment and selection of total crown cubic volume. μCT scans were aligned using three points in the first molar: 1—the mesial apical constriction, 2—the distal apical constriction, 3—coronal part of the mesial canal. Total crown volume from the mesial cemento-enamel junction was marked (white rectangle) over the slice in which the coronal pulp, radicular pulp of both canals, and both apical foramens are presented. Total crown cubic volume (mm$^3$) was determined by the marked rectangle over all slices in which crown enamel was observed.

The mean volumetric percentage of 500–1500 mgHA/ccm range of each group, which represent demineralized tissue, was compared.

### 2.8. Viable Count Evaluation

A low-speed dental bur 0.25 mm (MDT, Afula, Israel) was used to collect a hard tissue surface sample from the third molar. The bur was placed in an Eppendorf tube with 100 μL PBS and sonicated for 5 min to disrupt the biofilm (Bandelin sonopuls HD 2200, Berlin, Germany). Then, the tube was vortexed and plated on mitis salivarius agar plates (Difco, Detroit, MI, USA) for live bacterial count evaluation (CFU/mL).

### 2.9. Caries Scoring Using a Scoring Method over Hemi-Sectioned Molars

Caries lesions were evaluated using Keyes' caries scoring method with slight modification [14].

First, mandibular molars were hemi-sectioned along the mesiodistal sagittal plane using Super-Snap finishing and polishing disks (SHOFU inc., Kyoto, Japan).

Hemi-sectioned molars were photographed using an SMZ25 stereoscope (Nikon, Tokyo, Japan). The occlusal surface of the first molars was divided into eight surfaces; a vertical tangent for the tip of each cusp and the depth of each fissure was marked. The verticals divided the occlusal surface into eight units (as shown in Figure 3). Lesions were scored as the number of occlusal surfaces that were diagnosed with caries that have reached the cemento-enamel junction (CEJ).

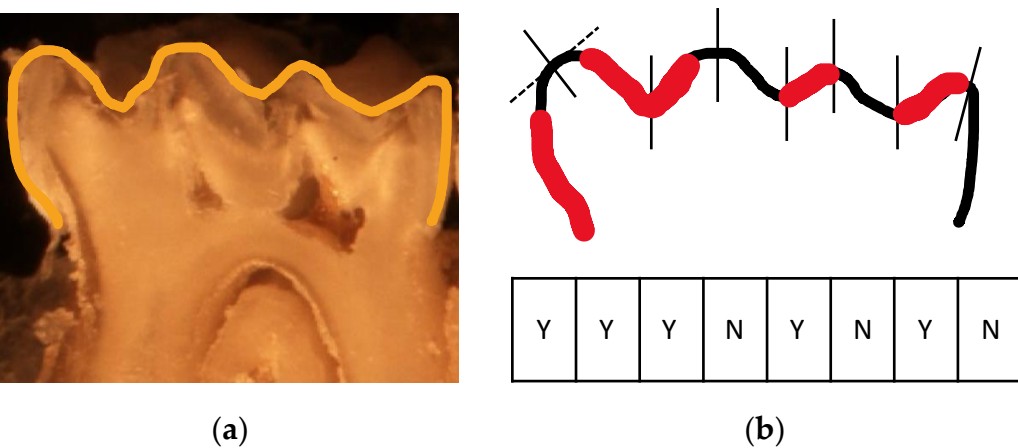

(**a**) (**b**)

**Figure 3. Caries scoring using hemi-sectioned molars.** First mandibular molars were hemi-sectioned along the mesiodistal plane and photographed using an SMZ25 stereoscope. (**a**) Representative hemi-section of caries-infected mandibular molar with marked occlusal surface outline. (**b**) Occlusal surface was divided into eight surfaces; a vertical tangent for the tip of each cusp and the depth of each fissure was marked. The verticals divided the occlusal surface into eight units. Each unit was marked using the table below: Y = Caries was detected, N = No caries was detected. The units identified as caries infected were marked in red and were summed to numeric score.

### 2.10. Statistical Analysis

The results were analyzed as the mean $\pm$ standard error of the mean of 5 samples in each experimental group. Statistical significance was calculated by Student's *t*-test (significance level: $p < 0.01$).

### 2.11. In Vivo Dental Caries Promotion

2.11.1. Mice

Caries was induced according to Keyes et al. [13] with modifications, using mice rather than rats.

Fifteen BALB/c female mice (6–7 weeks old) were purchased from Harlan (Jerusalem, Israel). All the animals were housed in ventilated cages at room temperature under a 16 h light and 8 h dark cycle. Ten mice received 10% sucrose containing distilled water and cariogenic diet KEYES #2000 [13] ad libitum, while five mice served as a control group receiving distilled water and normal diet ad libitum. All animal experimental procedures were reviewed and approved by the IACUC of the Hadassah-Hebrew University Medical Center (MD-17-15315-3).

A group of five mice was subjected to bacterial infection; five mice received high-sucrose diet only while five mice served as a control group.

The time allotted for the experiment is 42 days, as in previous studies [10,16,17]; the lesions developed within 5–7 weeks.

2.11.2. Bacterial Infection

*S. mutans UA159* was grown and adjusted to total bacterial load of $10^8$ (CFU/mL) similarly as described above for in vitro experiments. Bacterial infection was performed every 48 h, using a 0.2 mL oral gavage of bacterial suspension with 2% carboxymethyl cellulose (CMC).

### 2.11.3. Bacterial Outgrowth Evaluation and Carious Lesions Assessment

After 42 days, microbiological samples were collected using an oral swab from the murine mouths for 20 s. Viable bacterial number was evaluated (CFU/ mL) on mitis salivarius for *S. mutans* and BHI agar plates for total bacterial count.

The experiment was terminated after 42 days. Mice were anesthetized and euthanized. Hemi-mandibula were harvested. All tests were carried similarly as in the in vitro experiments. The hemi-mandibula were photographed using an SMZ25 stereoscope (Nikon, Tokyo, Japan) and scanned using micro-computed tomography (μCT) (μCT 40, Scanco Medical, Switzerland) for demineralization quantification.

## 3. Results

### *3.1. Assessment of Caries Lesions in Extracted—Jaws from Dissected Healthy Mice*

### 3.1.1. Clinical Evaluation

Caries lesions were detected clinically in the *UA159* bacterial infected group within 5 days. Representative images of an average lesion developed, before, and after bacterial infection, are shown in Figure 4(B1,B2), respectively. No carious lesions were observed in the non-infected hemi-mandibles, as shown in Figure 4(A1,A2).

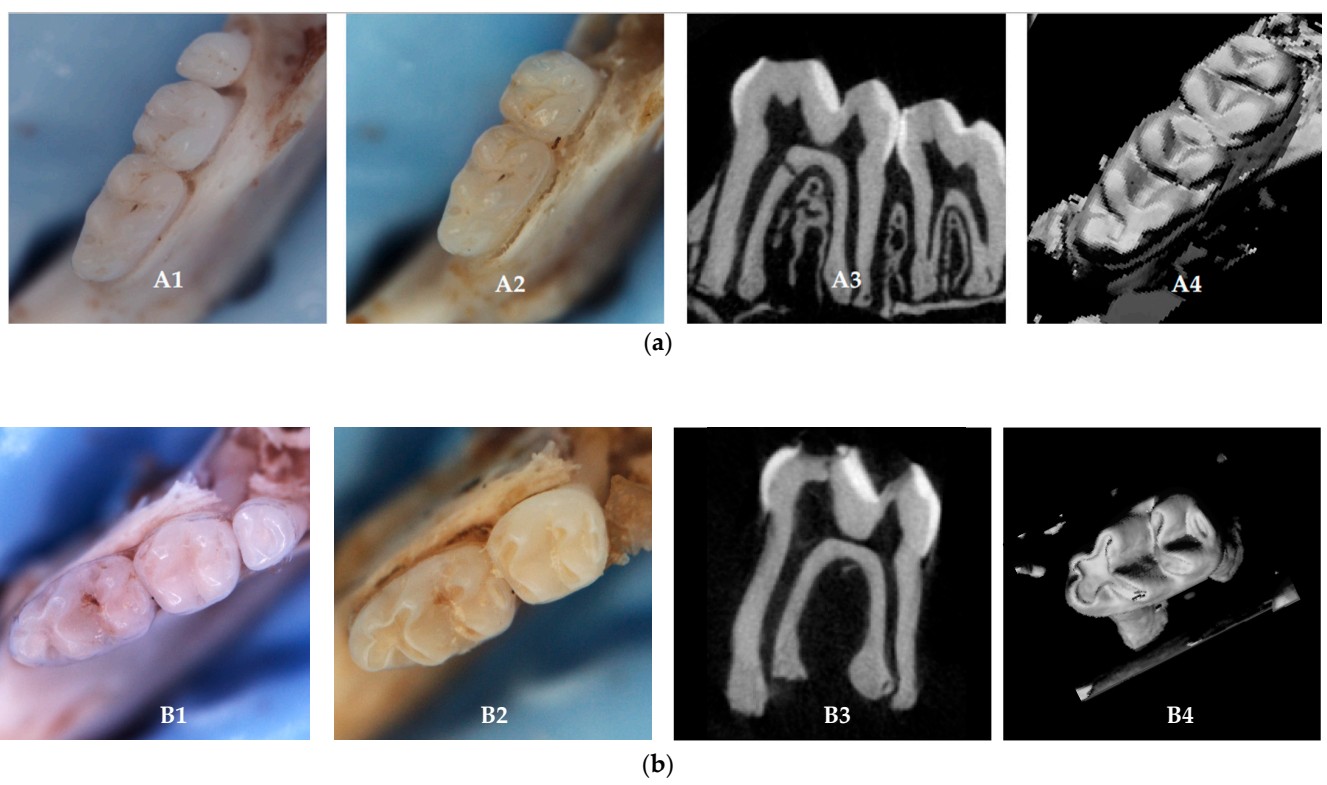

**Figure 4. Evaluation of caries lesions, clinical and radiographic, after 5 days in a caries-promoting environment**. Representative clinical and radiographic images. (**a**) represents the no bacteria group; (**b**) represents the *S. mutans* UA159 bacterial-infected group. Clinical photos were depicted before (**A1,B1**) and after (**A2,B2**) caries induction using high-sucrose media and cariogenic bacteria (*S. mutans UA159*) for 5 days. Radiographic (**A3,A4**) scans were used for demineralization evaluation and quantification; hemi-mandibles were segmented and reconstructed to acquire 3D images by the μCT (**A4,B4**).

### 3.1.2. Quantification of Demineralization

μCT scans demonstrated carious lesions in the bacterial-infected groups, while teeth in the control group remain intact, depending on the clinical appearance. Representative radiographic images after infection are shown in Figure 4(A3,B3).

In the *UA159* bacterial-infected group, the radiographic images demonstrate extensive dentin lesions in fissures (Figure 4(B3)), while the teeth in the control group remain intact (Figure 4(B4)).

Figure 5 shows the differences in the caries range before and after inducing caries-promoting conditions. A significant increase was observed in the bacterial-infected group ($p < 0.0001$).

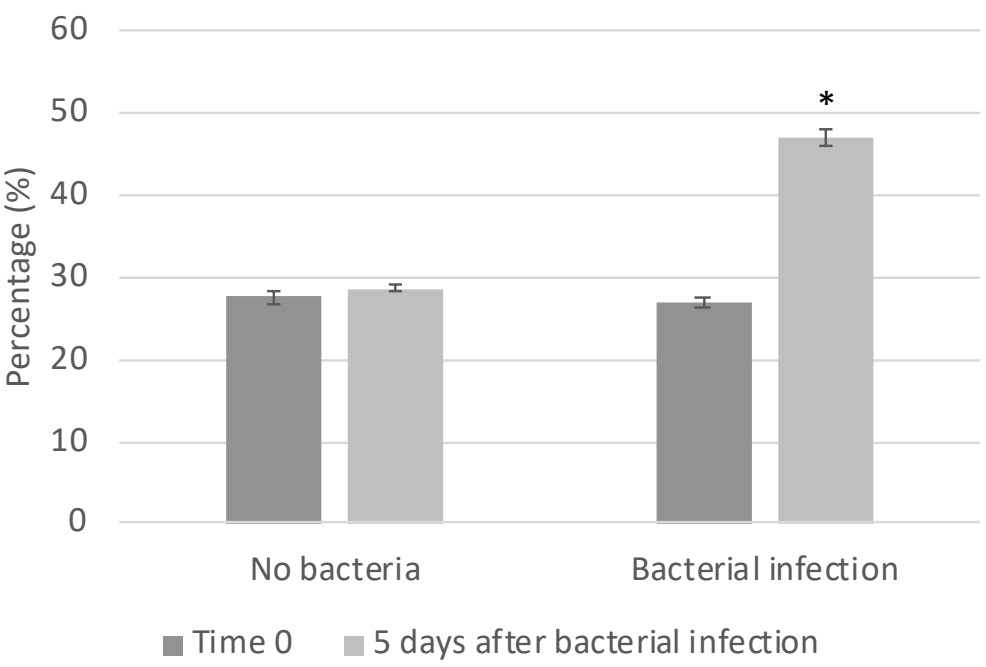

*p<0.01

**Figure 5. Quantification of demineralization.** μCT analysis comparison of volumetric percentage in the pulp and caries representing range (500–1500 mgHA/ccm) before (dark gray) and after (light gray) the experimental period. A significant increase ($p = 0.000069$) in the caries representing range was observed in the *S. mutans UA159* bacterial-infected group.

### 3.1.3. Bacterial Outgrowth Evaluation

At the end-point of the experiments, bacterial viable count in the bacterial-infected group was around $10^4$ CFU/mL, while the control group remained non-contaminated.

### 3.1.4. Caries Scoring over Hemi-Sectioned Molars

As shown in Figure 6, the caries score over hemi-sectioned molars increased significantly ($p = 0.011$) in the bacterial infected jaws (n = 5) compared to the no bacteria jaws (n = 5). Figure 2a,b show representative images of hemi-sectioned molars from the bacterial-infected group with a caries score equal to 5 (units out of 8).

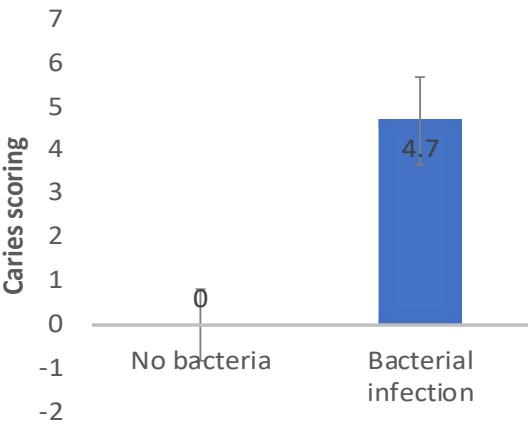

**Figure 6. Caries scoring over hemi-sectioned molars after 5 days in caries-promoting conditions.** Caries score increased significantly ($p = 0.011$) in the bacterial-infected jaws (n = 5) compared to the no bacteria jaws (n = 5).

*3.2. Assessment of Caries Lesions in Jaws from Experimental In Vivo Caries Model in Mice*

3.2.1. Clinical Evaluation

Clinical photographs demonstrated carious lesions development in the *UA159* bacterial-infected group, while the teeth in the control and high-sucrose diet groups remained intact. Representative photographs shown in Figure 7a.

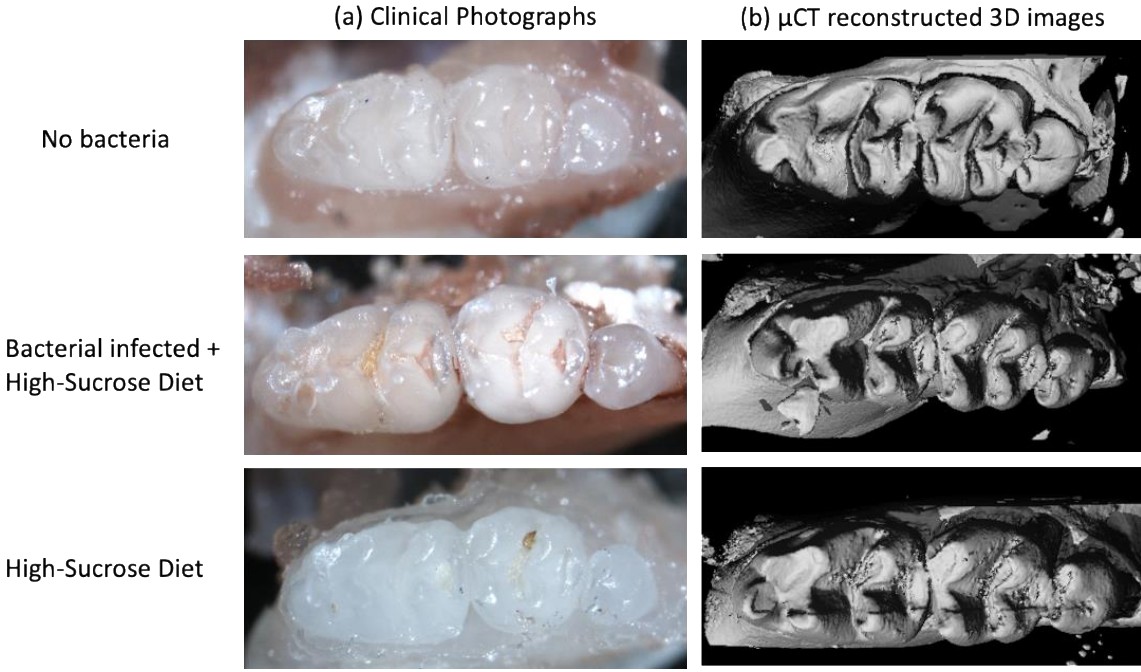

**Figure 7.** Clinical and radiographic evaluation demonstrates carious lesions development by *S. mutans UA159* bacterial infection and high-sucrose diet. (**a**) Hemi-mandibles were photographed using stereomicroscope. Representative clinical photographs depict extensive carious lesions developed in the *S. mutans UA159* bacterial infected + high-sucrose diet group, no carious lesions were depicted in the control and high-sucrose diet groups. (**b**) Hemi-mandibles were segmented and reconstructed to acquire 3D images by the μCT.

3.2.2. Quantification of Demineralization

μCT radiograph scans demonstrated extensive carious lesions in the *UA159* bacterial-infected group, while for the teeth in the control and high-sucrose diet groups, no demineralization was shown. Representative radiographic images are shown in Figure 8b–d.

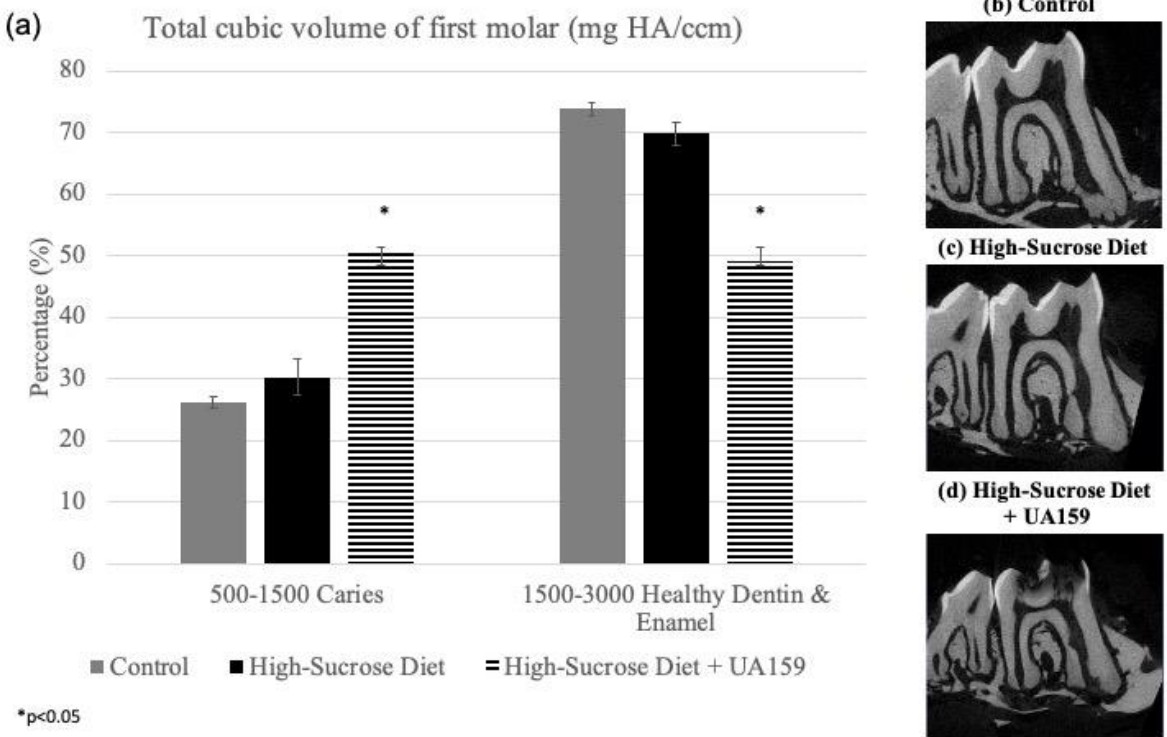

**Figure 8. Radiographic μCT analysis shows carious lesions induced by high-sucrose diet and *UA159 S. mutans*.** μCT radiographs were analyzed using a dedicated algorithm to divide total crown volume into density ranges. (**a**) Comparison of the volumetric percentage of density ranges by mgHA/ccm (i.e., caries representing range was determined to 500–1500 mgHA/ccm, healthy dentin and enamel representing range was determined to 1500–3000 mgHA/ccm). Mice were divided into three groups: high-sucrose diet (Keyes' #2000 and 10% sucrose containing distilled water) with *UA159* bacterial-infected group (n = 5) (striped column) compared to control (receiving normal diet and distilled water, n = 5) (gray column) and high-sucrose diet (without bacterial infection, n = 5) (black column). Representative radiographic images of the first molar from each group: (**b**) control, (**c**) high-sucrose diet, and (**d**) high-sucrose diet + *UA159*.

When comparing the total cubic volume of the first molar's crown divided into six selected ranges of density (by mgHA/ccm), a significant increase ($p < 0.05$) in the volumetric percentage of the density representing pulp and carious dentin (500–1500 mgHA/ccm) was observed (Figure 8a).

As shown in Figure 7b, μCT radiograph scans were reconstructed by the μCT to acquire 3D images. The 3D reconstructed images demonstrate and emphasize clinical appearance.

### 3.2.3. *S. Mutans* Dominates Oral Habitat after Infection

After 6 weeks of treatment, cariogenic diet induced an increase of *S. mutans* bacterial load. As expected, higher and significant increase (≈4 log increase) was shown in the bacterial infection group, compared to the control group. Statistical significance was calculated by Student's *t*-test (significance level: $p < 0.05$) compared to the untreated control (n = 5 in every tested group).

Respectively, an Increase in the relative portion of *S. mutans* from total bacteria count was shown in the high-sucrose diet and bacterial-infected group (Figure 9). Total bacteria viable count was evaluated using dilutions plated on BHI agar plate, while *S. mutans* viable count was evaluated using dilutions plated on a mitis–salivarius–bacitracin agar plate. The results are presented as percentages normalized to the total bacteria count in each group.

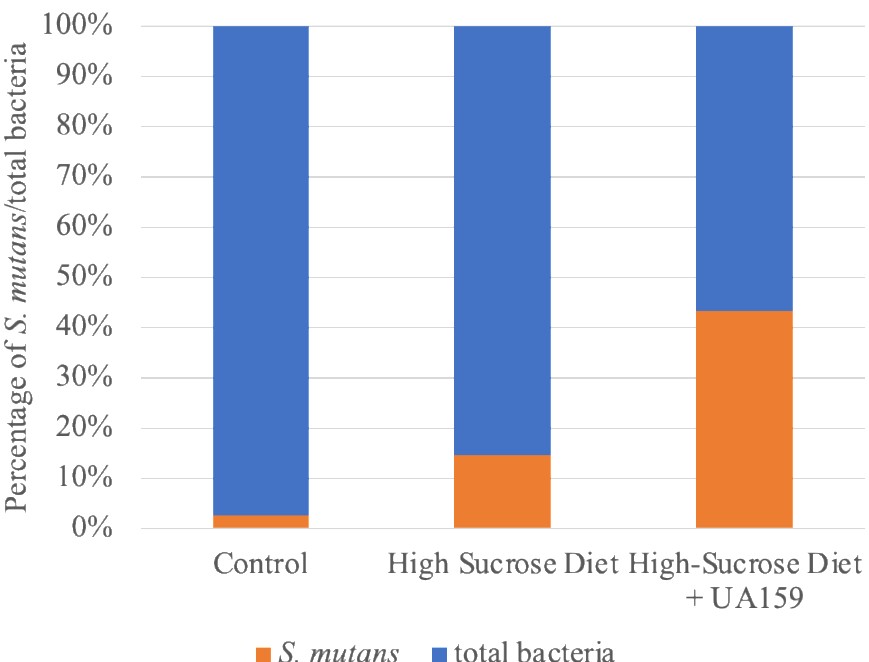

**Figure 9.** Cariogenic diet and bacterial infection increase the relative percentage of *S. mutans* from total bacterial counts in the mice oral habitat. Cariogenic (high-sucrose) diet and bacterial infections induced a significant increase of *S. mutans* bacterial load. The results are presented as percentages normalized to the total bacterial count in each group. Blue = total bacteria viable count. Orange =*S. mutans* viable count.

## 4. Discussion

An innovative in vitro caries model was developed herein aiming to decrease unnecessary animal sacrifice. Caries-promoting conditions mimicking in vivo caries models in murine were used to induce reproducible carious lesions within 5 days in extracted jaws of healthy animals sacrificed for other experiments and were no longer needed. To ensure the reliability of the in vitro development process, a resembling in vivo model study was conducted, and carious lesions were compared using qualitative and quantitative methods in both models. In the in vitro model, reproducible caries lesions were induced six times faster than in the in vivo model without compromising the data that can be drawn from the experimental conditions and without additional animal death.

Caries lesions treatment requires the removal of the infected hard dental tissues, and restoring the lost tissues using different types of restorative dental materials, mainly by amalgam, composite resin or glass-ionomer cement. Caries removal was mostly done by traditional low-speed and high-speed rotatory instruments. Recently, several innovative techniques have been suggested for minimal-invasive caries removal: sonic and ultrasonic oscillating devices [22], lasers [23], and chemo-mechanical solutions [24]. Caries modeling is necessary for evaluating innovative restorative and removal materials before clinical trials.

Maske et al. [25] have reviewed in vitro biofilm models to study dental caries. They described the high variability and complexity of in vitro models; numerous models were conducted over human teeth or enamel slabs. Demineralization is very difficult to diagnose and quantify using these models, and it often requires unique equipment with a highly sensitive technique, such as polarized light microscopy [26,27], microhardness [28,29], atomic force microscopy [30], or quantitative light fluorescence (QLF) [31,32].

Numerous caries studies have tried to quantify carious lesions development in vivo. Keyes [14] was the first to describe caries-scoring over hemi-sections rat molars. His scoring method, using murexide dye, is a common method still in use [33–35]. Zhang et al. [34] described the radiographic analysis of caries induced in rats; µCT scans were used to compare the mineral density of rats' molar enamel after calibration with hydroxyapatite standards of appropriate density. To our knowledge, Culp et al. [15] described exclusively the use of

Larson's modification [36] for Keyes' caries scoring method in a murine mice model; all mandibular and maxillary molars were scored for smooth surface and sulcal caries.

The clinical and radiographic demonstrations of the carious lesions evolved in the models described in the present study enables the identification of the lesion, quantification of the demineralization, and characterization in a similarity to the identification of carious lesions by dentists. Furthermore, the methods described require only stereomicroscope and µCT, the analysis is relatively simple, and the results are repeatable.

The demineralization process is dependent on the surface width and morphology. Murine molar teeth morphology contain sulci and fissures [37], which mimics the human molar morphology, but they are smaller, and the dental hard tissues are thinner than humans. Hence, the caries lesion development in the murine teeth is accelerated compared to the process in human teeth or enamel slabs [38]. For example, Park et al. [31] showed initial caries lesions over human molar enamel blocks after 30 days in caries-promoting conditions. The relatively rapid caries production in our model, after 5 days only, allows multiple repetitions, multiple dental materials evaluation, and avoidance of external infections or influences, and therefore provides a superior alternative over other in vitro models. Moreover, the use of extracted jaws from dissected healthy mice allowed us the control of multiple criteria, such as identical genetics, age, diet, growth conditions, and the starting condition of the dental hard tissues. Thus, monitoring multiple factors that may affect outcomes allows the examination and isolation of each factor and better understanding of carious lesions development and means to prevent it.

The mouth is a dynamic environment; salivary flow and its components have a protective role in reducing caries incidence [15,16,39]. The absence of salivary flow in the in vitro caries model may be the reason for more extensive lesion development, in shorter time, compared to the in vivo model. Moreover, clinical and radiographic photos exhibit some differences in the demineralization pattern: in the in vivo model, carious lesions were developed mostly in fissures' depth, while in the vitro model, enamel was detached, and demineralization occurred also in smooth surfaces.

In addition to animal lives, diet, and maintenance cost, the animal model requires a much longer time and effort compared to in vitro modeling. Given that the clinical and µCT analysis in both models are compatible and consistent, the in vitro model may offer many advantages over the in vivo model.

While animal experiments require skilled manpower, time-consuming protocols, and high cost [18], and in vitro acid-generated demineralization does not mimic the bacterial-induction of caries lesions in vivo, compared to other in vitro models, our in vitro model using murine extracted jaws, which mimics a natural caries-inducing environment, serves as a reliable and reproducible in vitro caries model.

## 5. Conclusions

A novel in vitro model using jaws from euthanized healthy mice was introduced. The present in vitro model allows reproducible extensive carious lesions within 5 days, similar to carious lesions that are induced within 6–7 weeks in vivo. The in vitro model may be beneficial for the evaluation of caries prevention and treatment means. It can be suggested that the model presented here may help reduce and replace animal use for caries research in the future.

**Author Contributions:** Conceptualization, Y.H.-H. and N.B.; methodology, Y.H.-H. and N.B.; validation, A.W.-Z., Y.H.-H. and N.B.; formal analysis, A.W.-Z.; investigation, A.W.-Z., Y.H.-H. and N.B.; resources, Y.H.-H. and N.B.; data curation, A.W.-Z., Y.H.-H. and N.B.; writing—original draft preparation, A.W.-Z.; writing—review and editing, A.W.-Z., Y.H.-H. and N.B.; visualization, Y.H.-H. and N.B.; supervision, Y.H.-H. and N.B.; project administration, Y.H.-H. and N.B.; funding acquisition, Y.H.-H. and N.B. All authors have read and agreed to the published version of the manuscript.

**Funding:** The research was supported from a grant from The Israel Science Foundation (ISF), grant number 986/16.

**Institutional Review Board Statement:** The study was conducted according to the guidelines of the Declaration of Helsinki and approved by the authorized by the institutional ethics committee #HMO-0706-16. All animal experimental procedures were reviewed and approved by the IACUC of the Hadassah—Hebrew University Medical Center (MD-17-15315-3).

**Informed Consent Statement:** Informed consent was obtained from all subjects involved in saliva collection for the study.

**Acknowledgments:** We would like to thank all those who contributed to this research, including Judith Goldstein, Yael Feinstein-Rotkopf from the Light Microscopy Laboratory of the intradepartmental unit of the Hebrew University, Raphael lieber for helping with μCT. In addition, we would like to thank David Polak and Joseph Tam for their help with euthanized healthy mice.

**Conflicts of Interest:** The authors declare no conflict of interest. The funders had no role in the design of the study; in the collection, analyses, or interpretation of data; in the writing of the manuscript, or in the decision to publish the results.

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
