# Peer review of "A Novel Dental Caries Model Replacing, Refining, and Reducing Animal Sacrifice"

_applsci, doi:10.3390/app11157141_

Round 1
Reviewer 1 Report
There are some weaknesses through the manuscript which need improvement. Therefore, the submitted manuscript cannot be accepted for publication in this form, but it has a chance of acceptance after a minor revision. My comments and suggestions are as follows:
1- Abstract gives information on the main feature of the performed study, but some details about the experimental procedure must be added.
2- Authors must clarify necessity of the performed research.
3- The literature study must be enriched. In this respect, authors must read and refer to the following papers: (a) dental composite: https://doi.org/10.1016/j.jmbbm.2019.02.009 (b) fracture in dental resin: https://doi.org/10.1016/j.dental.2018.11.023 The current version of introduction is short.
4- It is a necessity to add some figures to show concept and some conditions. For instance, in section 2.
5- The main reference of each formula must be cited. Moreover, each parameters in equations must be introduced. Please double check this issue.
6- A scale bar must be added to Fig. 1.
7- statistical analysis must be presented in more detail and in more scientific way.
8- Reason for the standard deviation is the presented curves must be discussed. In addition, error in calculation must be considered and discussed.
9- In its language layer, the manuscript should be considered for English language editing. There are sentences which have to be rewritten.
10- The conclusion must be more than just a summary of the manuscript. It is too short. List of references must be updated based on the proposed papers. Please provide all changes by red color in the revised version.
Author Response
There are some weaknesses through the manuscript which need improvement. Therefore, the submitted manuscript cannot be accepted for publication in this form, but it has a chance of acceptance after a minor revision. My comments and suggestions are as follows:
- Abstract gives information on the ain feature of the performed study, but some details about the experimental procedure must be added.
- Experimental procedure is described in the abstract section as follows:
“hemi-mandibles were extracted from previously euthanized healthy 10-weeks old BALB/C female mice. Jaws were subjected to saliva, high sucrose diet and dental caries bacteria Streptococcus mutans UA159 for 5 days. Similar caries induction protocol was used in vivo in fifteen BALB/c female mice (6–7 -week-old) and compared to the in vitro model. Caries lesions were assessed: clinically by photographic analysis, by µCT analysis and bacterial growth was evaluated”
Further elaboration can be found in the materials and method section.
- Authors must clarify necessity of the performed research.
- In the introduction section, the necessity of caries models is described as follows:
“Caries models are valuable when trying to understand the complex process of the disease by addressing various questions related to biofilm formation, the process of caries development, and its prevention. Moreover, caries models are extremely valuable to determine prevention and treatment approaches prior to clinical application. Various in vitro and in vivo techniques have been developed for caries simulation. Many of these studies focus on the evaluation of bacterial growth or biofilm composition rather than demineralization of the enamel [6-9] . Furthermore, most in vivo caries studies have been conducted using rats rather than mice and facilitated cariogenic bacteria by including oral shifting via desalivation and/or antibiotic treatment which created a non-natural environment [10-15].”
- The literature study must be enriched. In this respect, authors must read and refer to the following papers: (a) dental composite: https://doi.org/10.1016/j.jmbbm.2019.02.009 (b) fracture in dental resin: https://doi.org/10.1016/j.dental.2018.11.023 The current version of introduction is short.
- We were unable to understand how to refer our caries models to the papers that were suggested. The suggestion was unclear. We will be glad to add the references and kindly ask for the reviewers' clarification.
- It is a necessity to add some figures to show concept and some conditions. For instance, in section 2.
- Thank you for this suggestion. A schematic diagram to summarize the in vitro model described was added- Figure 1.
- The main reference of each formula must be cited. Moreover, each parameters in equations must be introduced. Please double check this issue.
- We would be glad to add any missing information. The models that were introduced in the manuscript were evaluated for caries-modeling without examining any formulas.
- A scale bar must be added to Fig. 1.
- Good point. A scale bar was added to Fig. 1, which was changed to Fig. 2.
- statistical analysis must be presented in more detail and in more scientific way.
- Statistical analysis can be found in section 2.10:
“2.10 Statistical analysis
The results were analyzed as the mean ± standard error of the mean of 5 samples in each experimental group. Statistical significance was calculated by Student’s t-test (significance level: p < 0.05).”
- Reason for the standard deviation is the presented curves must be discussed. In addition, error in calculation must be considered and discussed.
Comparison between experimental groups, which contains 5 sample in each group, was analyzed by student’s t-test. Significance level was determined to p<0.05). standard deviation was evaluated to verify the dispersion in not too broad.
- In its language layer, the manuscript should be considered for English language editing. There are sentences which have to be rewritten.
- Manuscript had undergone English language editing by Wiley Editing Service prior to submission. A certificate can be supplied upon request.
10- The conclusion must be more than just a summary of the manuscript. It is too short. List of references must be updated based on the proposed papers. Please provide all changes by red color in the revised version.
- we elaborated the conclusion section as follows:
“A novel in vitro model using jaws from euthanized healthy mice was introduced. The present in vitro model allows reproducible extensive carious-lesions within 5 days, similar to carious-lesions that are induced within 6-7 weeks in vivo. The in vitro model may be beneficial for evaluation of caries prevention and treatment means. It can be suggested that the model presented here may help reduce and replace animal use for caries research in the future. “
- The References were updated and reference section accordingly. References that were added:
- Culp, D.J., B. Robinson, and M.N. Cash, Murine Salivary Amylase Protects Against. Front Physiol, 2021. 12: p. 699104.
- Basir, L., et al., Anticaries Activity of Curcumin on Decay Process in Human Tooth Enamel Samples (In Vitro Study). J Natl Med Assoc, 2018. 110(5): p. 486-490.
- Giacaman, R.A., P. Jobet-Vila, and C. Muñoz-Sandoval, Anticaries activity of egg ovalbumin in an experimental caries biofilm model on enamel and dentin. Clin Oral Investig, 2019. 23(9): p. 3509-3516.
- Castro, R.J., et al., Anti-caries effect of fluoridated milk-based drink consumed by older adults on an in vitro root caries experimental model. Arch Oral Biol, 2020. 118: p. 104878.
- Cardoso, M., et al., Efficacy and Patient's Acceptance of Alternative Methods for Caries Removal-a Systematic Review. J Clin Med, 2020. 9(11).
- Luo, J., et al., Novel lactotransferrin-derived synthetic peptides suppress cariogenic bacteria. J Oral Microbiol, 2021. 13(1): p. 1943999.
- Zhang, Q., et al., Inhibitory Effect of Lactobacillus planetarium CCFM8724 towards Streptococcus mutans- and Candida albicans- Induced Caries in Rats. Oxid Med Cell Longev, 2020. 2020: p. 4345804.
- Liang, J., et al., Effects of a derivative of reutericin 6 and gassericin A on the biofilm of Streptococcus mutans in vitro and caries prevention in vivo.Odontology, 2021. 109(1): p. 53-66.
Reviewer 2 Report
This is a very interesting study on a new method of simulating the formation of carious lesions using in vitro models compared to the classic in vivo model.
I appreciate the authors' attempt but the whole system must be validated at the level of literature
For this work some criticisms are present:
Insert at the end of the introduction section the null hypotheses of the study which must then be refuted in the light of the results obtained
-The acceptance of the Ethics Committee should be included in an initial paragraph of the materials and methods
-What studies have been followed to replicate the cariogenic situation? Indicate them explicitly in the materials and methods section
-In the title of paragraph 2.8 do not indicate that it is a new method but describe it in the written part of the paragraph itself
- sub-paragraphs 2.10.3 and following must be combined in a single paragraph, modifying the single title
-I believe that other studies are needed to validate the quantification of carious lesions proposed by the authors; in this sense it is necessary to report in the discussion section other studies on this aspect present in the literature
-Also in the discussion section, in its initial part, the systems for removing carious lesions should be summarized, from the most classic to the most innovative. In this regard, I recommend that you insert the following scientific work in the reference section which could be of help:
Cianetti S, Abraha I, Pagano S, Lupatelli E, Lombardo G. Sonic and ultrasonic oscillating devices for the management of pain and dental fear in children or adolescents that require caries removal: a systematic review. BMJ Open. 2018 Apr 28; 8 (4): e020840. doi: 10.1136 / bmjopen-2017-020840. PMID: 29705764; PMCID: PMC5931288.
-The reference section appears very dated. I ask the authors for an effort of innovation on the search for newer works
Author Response
This is a very interesting study on a new method of simulating the formation of carious lesions using in vitro models compared to the classic in vivo model.
I appreciate the authors' attempt but the whole system must be validated at the level of literature
- Thank you
For this work some criticisms are present:
- Insert at the end of the introduction section the null hypotheses of the study which must then be refuted in the light of the results obtained
- We have inserted the null hypothesis at the end of the introduction section:
“The null hypotheses of this study were that development of an in vitro caries model using extracted jaws from euthanized healthy mice, will mimic the common in vivo caries model, and will introduce an alternative approach for dental research experiments. “
-The acceptance of the Ethics Committee should be included in an initial paragraph of the materials and methods
- Thank you for this comment. An ethics approval paragraph 2.1 was added in initial paragraph of the materials and methods section :
“2.1 Ethics approval
Saliva was collected from 5 healthy volunteers, age range 22-67 (3 female, 2 men) (authorized by the institutional ethics committee #HMO-0706-16). All participants gave verbal and written assent to participate.
All animal experimental procedures were reviewed and approved by the IACUC of the Hadassah—Hebrew University Medical Center (MD-17-15315-3).”
-What studies have been followed to replicate the cariogenic situation? Indicate them explicitly in the materials and methods section
- Description of the study to replicate the cariogenic situation was added. The following sentence was added to section 2.11.1:
“Caries promotion according to Keyes’ et al. [13] with modifications was conducted, using mice rather than rats.”
-In the title of paragraph 2.8 do not indicate that it is a new method but describe it in the written part of the paragraph itself
- The word “novel” was removed from the paragraph’s title.
- The following paragraph 2.8 section after rewriting:
“Caries lesions were evaluated using Keyes’ caries scoring method with slight modification [14].
First mandibular molars were hemi-sectioned along mesiodistal sagittal plane using Super-Snap finishing and polishing disks (SHOFU inc., Kyoto, Japan).
Hemi-sectioned molars were photographed using SMZ25 stereoscope (Nikon, Tokyo, Japan). Occlusal surface of first molars was divided into eight surfaces; a vertical tangent for the tip of each cusp and the depth of each fissure was marked. The verticals divided the occlusal surface into eight units (as shown in Fig. 3). Lesions were scored as number of occlusal surfaces that were diagnosed with caries that has reached the Cemento-Enamel Junction (CEJ)”
- sub-paragraphs 2.10.3 and following must be combined in a single paragraph, modifying the single title
- Paragraphs 2.10.4 was combined with paragraph 2.10.3 to the single title “Bacterial Outgrowth Evaluation and Carious lesions assessment”, paragraph’s number was changed to 2.11.3
-I believe that other studies are needed to validate the quantification of carious lesions proposed by the authors; in this sense it is necessary to report in the discussion section other studies on this aspect present in the literature
- we elaborated the discussion section about current studies quantifying carious lesion. We have added the following paragraph:
“Numerous caries studies have tried to quantify carious-lesions development in vivo. Keyes[14] was the first to describe caries-scoring over hemi-sections rat molars. His scoring method, using murexide dye, is a common method still in use[34-36]. Zhang et al.[35]described radiographic analysis of caries induced in rats; µCT scans were used to compare mineral density of rats’ molar enamel after calibration with hydroxyapatite standards of appropriate density. To our knowledge, Culp et al.[15] described exclusively the use of Larson’s modification [37] for Keyes’ caries scoring method in murine mice model; all mandibular and maxillary molars were scored for smooth surface and sulcal caries.”
-Also in the discussion section, in its initial part, the systems for removing carious lesions should be summarized, from the most classic to the most innovative. In this regard, I recommend that you insert the following scientific work in the reference section which could be of help:
Cianetti S, Abraha I, Pagano S, Lupatelli E, Lombardo G. Sonic and ultrasonic oscillating devices for the management of pain and dental fear in children or adolescents that require caries removal: a systematic review. BMJ Open. 2018 Apr 28; 8 (4): e020840. Doi: 10.1136 / bmjopen-2017-020840. PMID: 29705764; PMCID: PMC5931288.
- Thank you for this suggestion. A paragraph summarizing caries-removal techniques was added in the discussion initial part including the suggested ref:
“Caries-lesions treatment requires the removal of the infected hard dental tissues, and restoring the lost tissues using different types of restorative dental materials, mainly by amalgam, composite resin or Glass-Ionomer-Cement. Caries removal mostly done by traditional low-speed and high-speed rotatory instruments. Recently, Several innovative techniques has been suggested for minimal-invasive caries removal; Sonic and ultrasonic oscillating devices [21], lasers [22] and chemo-mechanical solutions [23]. Caries modeling is necessary for evaluating innovative restorative and removal materials before clinical trials.”
-The reference section appears very dated. I ask the authors for an effort of innovation on the search for newer works
- The References were updated and reference section accordingly. References that were added:
- Culp, D.J., B. Robinson, and M.N. Cash, Murine Salivary Amylase Protects Against. Front Physiol, 2021. 12: p. 699104.
- Basir, L., et al., Anticaries Activity of Curcumin on Decay Process in Human Tooth Enamel Samples (In Vitro Study). J Natl Med Assoc, 2018. 110(5): p. 486-490.
- Giacaman, R.A., P. Jobet-Vila, and C. Muñoz-Sandoval, Anticaries activity of egg ovalbumin in an experimental caries biofilm model on enamel and dentin. Clin Oral Investig, 2019. 23(9): p. 3509-3516.
- Castro, R.J., et al., Anti-caries effect of fluoridated milk-based drink consumed by older adults on an in vitro root caries experimental model. Arch Oral Biol, 2020. 118: p. 104878.
- Cardoso, M., et al., Efficacy and Patient's Acceptance of Alternative Methods for Caries Removal-a Systematic Review. J Clin Med, 2020. 9(11).
- Luo, J., et al., Novel lactotransferrin-derived synthetic peptides suppress cariogenic bacteria. J Oral Microbiol, 2021. 13(1): p. 1943999.
- Zhang, Q., et al., Inhibitory Effect of Lactobacillus planetarium CCFM8724 towards Streptococcus mutans- and Candida albicans- Induced Caries in Rats. Oxid Med Cell Longev, 2020. 2020: p. 4345804.
- Liang, J., et al., Effects of a derivative of reutericin 6 and gassericin A on the biofilm of Streptococcus mutans in vitro and caries prevention in vivo. Odontology, 2021. 109(1): p. 53-66.
Reviewer 3 Report
It is innovative to design an in vitro caries models, following points should be addressed before acceptance,
- Please include the number of healthy volunteers recruited for saliva collection. Such as the age range and gender
- Figure 3, why did the teeth color change from white before treatment to clear after treatment? Any discussion?
- Please move the ‘Fig 2c’ Bacterial outgrowth evaluation to ‘results’ section
- Please include 3D reconstructed images for the in vitro models.
- For the caries lesions induced in vitro showed in Figure 3 B2 and B4, it seems the enamel was detached from dentine and suspected demineralization happened on the cusp tips, which is quite different from the in vivo results showed in Figure 5 and 6, more discussion is required to evaluate the efficiency of this in vitro model.
- In addition, in vivo mouth is a dynamic environment with saliva exchanging while this in vitro model is placing teeth in a static environment, it should be considered and discussed
Author Response
It is innovative to design an in vitro caries models, following points should be addressed before acceptance,
- Please include the number of healthy volunteers recruited for saliva collection. Such as the age range and gender
- Number, age range and gender were added in paragraph 2.1:
“Saliva was collected from 5 healthy volunteers, age range 22-67 (3 female, 2 men)”
- Figure 3, why did the teeth color change from white before treatment to clear after treatment? Any discussion?
- Thank you for your comment. The photographic depiction was replaced to avoid misunderstanding of the readers. The samples teeth color did not change, the light conditions of this specific photograph might mislead.
- Please move the ‘Fig 2c’ Bacterial outgrowth evaluation to ‘results’ section
- Fig 2c was moved to the results section and was change to Figure 6.
- Please include 3D reconstructed images for the in vitro models.
- 3D reconstructed images were added into Figure 4- A4+B4.
- For the caries lesions induced in vitro showed in Figure 3 B2 and B4, it seems the enamel was detached from dentine and suspected demineralization happened on the cusp tips, which is quite different from the in vivo results showed in Figure 5 and 6, more discussion is required to evaluate the efficiency of this in vitro model.
- The microCT depiction gap between enamel and dentin in the pictures is an artifact. We believe that if enamel detachment truly occurred it would result in exposed dentin in the interproximal areas, which clearly did not happen in any of the samples. to avoid misunderstanding, the picture was replaced.
- In addition, in vivo mouth is a dynamic environment with saliva exchanging while this in vitro model is placing teeth in a static environment, it should be considered and discussed
- That is a good point. The in vitro model presented in the manuscript is not a fully static environment. The growth media, which is the carbohydrate source, was replaced every 24 hours during the experimental period. Following the reviewers' comment we have added the following paragraph to the discussion section :
“The mouth is a dynamic environment; salivary flow and its components have a protective role in reducing caries incidence[15, 16, 36]. Absence of salivary flow in the in vitro caries model may be the reason for more extensive lesion development, in shorter time, compared to the in vivo model. Moreover, clinical and radiographic photos exhibit some differences in the demineralization pattern- in the in vivo model, carious lesions were developed mostly in fissures’ depth while in the vitro model, enamel was detached, and demineralization occurred also in smooth surfaces. “
Round 2
Reviewer 2 Report
all comments were added